# LEARNING ARGUMENTATIVE SUMMARIZATION WITH ITERATIVE REJECTION SAMPLING

## ABSTRACT

Summarization is a fundamental task for evaluating language understanding in both humans and machines, and serves as a crucial tool for information processing in our data-rich world. While Large Language Models (LLMs) have shown significant progress in summarization, they still struggle with domain-specific tasks such as zero-shot medical documentation, legal text, and argumentative summarization. To improve argumentative text understanding and summarization, we propose an iterative fine-tuning framework that trains LLMs on high-quality argument-summary pairs generated by the model itself. These pairs are filtered using similarity scores calculated by comparing reconstructed arguments from summaries with the original arguments, using rejection sampling, without external supervision. Our experiments demonstrate that this method improves argument summarization performance, achieving gains up to 11.88% in BERT F1 similarity scores between reconstructed and original arguments, over the vanilla model without such fine-tuning on a dataset of 200 r/ChangeMyView posts.

## 1 INTRODUCTION

Summarization, a task to perform on natural language on which unrelated or information of lesser importance is filtered out and only the core points are kept, is one of the human skills that LLMs are trained to simulate. A good grasp of summarization can showcase a solid understanding of the material and human language itself. However, research on Large Language Models (LLMs) has shown inconsistent performance across different domains. They have on-par performance when summarizing news, albeit in different styles (Zhang et al., 2024); when adapted to corresponding domains either by in-context learning or fine-tuning, they could perform even better than professionals on summarizing several types of clinical documents (Van Veen et al., 2024). However, when not adapted for the specific medical domains (Tang et al., 2023) or on legal (Deroy et al., 2024) and various long-form domains (Que et al., 2024), LLMs show significant performance degradation with inconsistencies and hallucinations that make them risky and unfit for real-world deployment.

The importance of LLMs to be good summarizers is multi-faceted. For one, the ability to summarize natural language written by humans could signify a solid understanding of the language, a fundamental goal for the language models (Zhao et al., 2025). LLMs, while initially finetuned on mass corpora from the internet on the task of causal language modeling, associate the definition and concept of summarizing with the behavior of extracting the cardinal information out of convoluted pieces of text. However, to make the language models eventually human-adjacent (OpenAI, 2023) in terms of lingual abilities, they should have comparable performance to human experts in all challenging specific domains. Practically, summarizing online media content, including arguments, is instrumental to moderation, analysis, and recommendation (Gurjar et al., 2025; Schluger et al., 2022; Lee et al., 2020; Bhatia et al., 2014; Hasan et al., 2025). For humans, LLMs can also lighten the cognitive load for absorbing the same information, reducing misunderstandings, and cutting down on time consumption by summarizing long or technical documents for people. This includes summarization for research papers (Langston & Ashford, 2024), legal (Deroy et al., 2024), clinical (Van Veen et al., 2024), and log data produced by automated systems, such as databases (Zhou et al., 2023), allowing people to absorb information and form decisions at faster rates than before.

When summarizing the human language, many aspects or characteristics could make the task extra difficult, for humans and automated systems alike. Text filled with complicated and delicate facts,

such as clinical and legal documents, obfuscates summarization as they induce factual errors for any human or automated system that does not understand the material well (Afzal et al., 2024). Documents with tighter-knit logic create points that are closer to the cardinal logic and other side information that has looser ties to that logic, requiring the summarizer to have a good grasp on the cardinal logic (Afzal et al., 2024). We could argue that the intricate-factual summarization tasks could be resolved by injecting required information with Retrieval-Augmented Generation (RAG) systems. Therefore, the challenge to better automated summarization is how we prepare these systems, which are often LLMs, so they could capture the cardinal logic of a given piece of natural language, for the evaluation of which we need one instance of language that only captures the logic-heavy nature and less of the factual side. Argumentative text presents particularly acute challenges for LLM summarization that go beyond the factual accuracy issues found in medical or legal domains. Unlike news articles or technical documents that follow predictable structures, argumentative discourse requires models to identify and preserve complex logical relationships, implicit premises, counterarguments, and rhetorical strategies constituting cores of persuasive reasoning.

Currently, LLMs face significant hurdles when summarizing argumentative text due to the inherent complexity and unique characteristics of such content. Argumentative dialogues often contain contradictory utterances with opposing viewpoints and logical relationships, making them substantially more convoluted to summarize compared to standard text (Zhao et al., 2023; Shakil et al., 2024). U-shaped positional bias, a phenomenon caused by the attention mechanisms of current LLMs where the LLM will place more attention to both ends of the text while ignoring the middle (Vaswani et al., 2017; Hsieh et al., 2024; Yi et al., 2025), was also pronounced and degraded the faithfulness in long-form argumentative summarizations (Elaraby & Litman, 2025a). LLMs also struggle with argument prioritization, as human-written summaries prefer to focus on core arguments while using supporting examples selectively, whereas LLMs try to list all arguments without distinguishing between central and trivial points (Fan et al., 2024). Finally, the performance of such summarization is not consistent across domains. Legal argument generation represents one of the most problematic domains, exposing three critical shortcomings: hallucination, inadequate abstention when arguments are untenable, and poor factor utilization where models fail to incorporate relevant factual elements from provided case materials into their generated arguments (Zhang & Ashley, 2025).

To address these limitations, we propose an iterative rejection sampling framework for improving argumentative summarization in Large Language Models. Our approach leverages the model's own generation capabilities to create high-quality training data through a cyclical process:

**Summary generation and reconstruction**: The model generates summaries of argumentative texts and expands these summaries back into full arguments.

**Rejection sampling**: We select training pairs using semantic similarity scoring to ensure quality, functioning without external supervision signals.

**Iterative fine-tuning**: This self-supervised approach prevents the common pitfall of iterative training systems that drift toward verbose, unfocused outputs.

We evaluate our method on one of the Reddit's r/ChangeMyView dataset [1], which provides a versatile combination of discourse that could help language models to become more generalizable across different argumentative tasks (Dutta et al., 2022). In addition, CMV provides an ideal testbed for argumentative summarization due to its authentic, naturally-occurring argumentative discourse where users present structured reasoning to challenge and defend viewpoints—capturing the complex logical relationships and persuasive strategies that make argument summarization particularly challenging for current LLMs (Syed et al., 2021; Jo et al., 2021; Qin et al., 2025; Elaraby & Litman, 2025b). Through iterative fine-tuning on progressively refined data that prioritizes concise yet comprehensive summaries, our method enables the model to better preserve core argumentative elements and logical flow without sacrificing the essential brevity that makes summaries useful.

---

[1]https://huggingface.co/datasets/agentlans/reddit-logic

## 2 RELATED WORKS

### 2.1 NEURAL SUMMARIZATION AND DOMAIN ADAPTATION

The field of neural summarization has been transformed by large-scale Transformer-based sequence-to-sequence models that employ diverse pre-training objectives tailored for text generation tasks (Wang et al., 2020; Raffel et al., 2019; Lewis et al., 2019). Despite these advances, domain adaptation remains a critical challenge for neural summarization systems. Early approaches that combined pre-trained encoders like BERT with decoders trained from scratch suffered from representation mismatches between the two components (Wang et al., 2020; Yang et al., 2019). While unified sequence-to-sequence pre-training approaches like BART, T5, and PEGASUS have addressed this architectural mismatch, significant performance gaps persist when these models are applied to specialized domains without proper adaptation (Wan & Bansal, 2022). Recent work has shown that domain-specific fine-tuning and careful data filtering can substantially improve performance, with filtered datasets leading to better results than training on noisy domain-specific data (Laskar et al., 2023; Duong et al., 2025). The challenge is particularly acute for specialized domains with highly technical terminology, complex argumentative structures, or factual requirements that differ from the general web text typically used in pre-training (Wan & Bansal, 2022).

### 2.2 ARGUMENTATIVE TEXT UNDERSTANDING

Neural approaches to argumentative text understanding have evolved from traditional rule-based methods to sophisticated transformer-based architectures that can identify and structure argumentative discourse. Recent work has shown that fine-tuning BERT for identifying argumentative units and relationships between them within a text and across texts provides a foundation for understanding complex argumentative structures (Fabbri et al., 2021). More advanced approaches now focus on end-to-end argument graph construction, where structured representations of claims and premises in argumentative text are built by connecting claim and premise argumentative discourse units (Fabbri et al., 2021). These graph-based representations capture the logical flow of arguments more effectively than linear text processing approaches, enabling better preservation of reasoning chains in downstream tasks like summarization. The challenge becomes particularly acute when dealing with debate datasets and persuasive text, where maintaining the integrity of argumentative structure is crucial for generating coherent and logical summaries that preserve the original reasoning patterns.

### 2.3 EVALUATION OF AUTOMATED SUMMARIES

The evaluation of neural summarization systems has evolved beyond lexical overlap metrics like ROUGE to incorporate semantic similarity measures that better capture summary quality. Current pre-training works in abstractive summarization give more points to the summaries with more words in common with the main text and pay less attention to the semantic similarity between generated sentences and the original document (Salemi et al., 2021). Recent approaches have shown that experimental results using state-of-the-art performance on summarization tasks measured by both ROUGE and BERTScore, with human evaluation demonstrating that using semantic scores significantly improves summarization results (Salemi et al., 2021). Automatic evaluation with BLEU, ROUGE, and METEOR can be complemented by human judges who rate system outputs for relevance and coherence, providing a more comprehensive quality assessment (Hua & Wang, 2020).

Faithfulness assessment has become a critical component of summarization evaluation, addressing the persistent issue that recent models can achieve highly fluent and coherent abstractive summaries, yet the generated summaries often contain factual errors (Wan & Bansal, 2022). Advanced evaluation frameworks now demonstrate that training methods can achieve up to a 14% improvement in faithfulness metrics over existing methods according to automatic evaluation metrics, with evaluations by both GPT-4 and human judges indicating substantially more faithful generations (Duong et al., 2025). Human evaluation protocols have also been refined to assess specific aspects of summary quality, with evaluators scoring factual consistency and coherence on scales that provide more nuanced assessment than binary judgments (Laskar et al., 2023). They enable better understanding of trade-offs between fluency, semantic fidelity, and factual accuracy in summarization systems.

### 2.4 POLICY OPTIMIZATION METHODS FOR LLM ALIGNMENT

Large Language Models require alignment with human preferences when deployed in real-world settings, leading to the development of various reward-based fine-tuning techniques (Feng et al., 2024). Proximal Policy Optimization (PPO) is a popular approach towards aligning LLMs to human preferences by reward models trained on human preference data (Stiennon et al., 2020). Unlike PPO, Group Relative Policy Optimization (GRPO) operates by evaluating advantages based on the relative performance of multiple sampled outputs for a given prompt, calculating advantages by normalizing rewards relative to group performance statistics such as mean and standard deviation (Kumar et al., 2025; Chen et al., 2025). Reward-ranked Fine-tuning (RAFT) offers a straightforward alternative to complex reinforcement learning approaches by operating through an iterative three-step framework: generating a batch of outputs from the model, scoring outputs using a reward function and filtering high-reward samples, and fine-tuning the model on the filtered data (Dong et al., 2023). Further research reveals that similar, improved rejection-sampling based approaches that remove both the fully correct and incorrect responses also work for GRPO better than the vanilla (Xiong et al., 2025).

## 3 METHODOLOGY

Many of the challenges outlined in Section 2 are imposed either by the limitations regarding the distribution of topics learned by LLMs, the mechanisms that current LLMs use, or the lack of domain-specific data for adaptation. These challenges, which might have caused the observed hallucination or omission of details, could be mitigated by adapting LLMs to these domains. However, the inability to address and extract the cardinal logic is a more universal flaw that must be addressed. To address the limitation where current LLMs struggle with argumentative summarization due to complex logical relationships and lack of domain-specific data, we propose an iterative rejection sampling framework that addresses these limitations through self-supervised fine-tuning of the model.

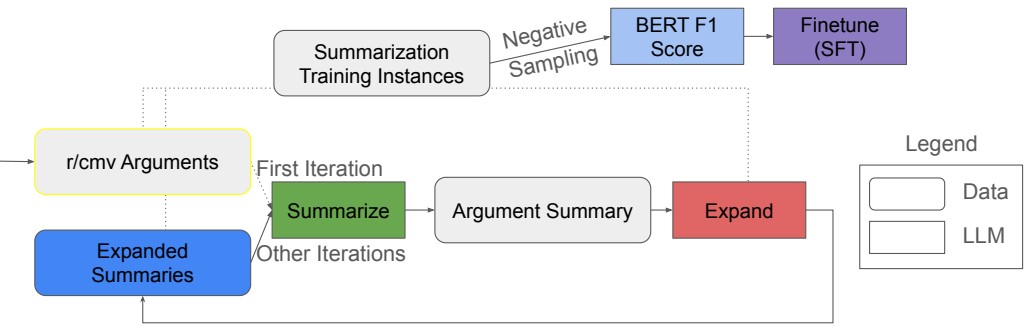

Figure 1: Overview of the iterative rejection sampling framework for argumentative summarization. The process cycles through summary generation, reconstruction, negative sampling, and model fine-tuning across multiple iterations. For each argument or an expanded summary, it was summarized while sampling multiple times, creating diverse summaries that are evaluated and ranked according to the similarity between their expanded summary versions and the original arguments.

### 3.1 PROBLEM FORMULATION

Let $\mathcal{D} = \{(a_i, s_i)\}_{i=1}^N$ be a dataset of argumentative text-summary pairs, where $a_i$ represents an argumentative text and $s_i$ represents its corresponding summary. Our goal is to learn a summarization function $f_\theta : \mathcal{A} \rightarrow \mathcal{S}$ parameterized by $\theta$, where $\mathcal{A}$ is the space of argumentative texts and $\mathcal{S}$ is the space of summaries. Given the scarcity of high-quality argumentative summarization data, we propose an iterative self-training framework that generates its own training data through reconstruction-based quality assessment. Let $g_\phi : \mathcal{S} \rightarrow \mathcal{A}$ denote a reconstruction function that expands summaries back to argumentative texts. The key insight is that high-quality summaries should enable faithful reconstruction of the original argumentative content.

### 3.2 ITERATIVE REJECTION SAMPLING FRAMEWORK

Our iterative rejection sampling framework operates through a cyclical process that generates high-quality argumentative text-summary pairs without requiring external supervision, consisting of: summary generation and reconstruction, rejection sampling, and iterative fine-tuning.

#### 3.2.1 SUMMARY GENERATION AND RECONSTRUCTION

At each iteration $t$, we begin with a set of argumentative texts $\mathcal{A}^{(t)} = \{a_1^{(t)}, a_2^{(t)}, \ldots, a_n^{(t)}\}$. For the initial iteration ($t = 0$), this corresponds to the original training corpus. The model $f_{\theta^{(t)}}$ generates $k$ summary candidates for each argumentative text:

$$S_i^{(t)} = \{s_{i,1}^{(t)}, s_{i,2}^{(t)}, \ldots, s_{i,k}^{(t)}\} = \{f_{\theta^{(t)}}(a_i^{(t)})\}_{j=1}^k \tag{1}$$

where each summary $s_{i,j}^{(t)}$ is sampled from the model's distribution with temperature $T = 1.5$ and minimum probability threshold $p_{min} = 0.1$.

Subsequently, we employ the same model to reconstruct argumentative texts from these summaries using an expansion function $g_{\theta^{(t)}}$:

$$\tilde{a}_{i,j}^{(t)} = g_{\theta^{(t)}}(s_{i,j}^{(t)}) \tag{2}$$

The reconstruction process uses the prompt template: "Expand the following summaries to one argumentative paragraph with 18 sentences that you might find on reddit's r/changemyview."

#### 3.2.2 QUALITY FILTERING VIA RECONSTRUCTION SIMILARITY

To select high-quality training pairs, we implement a rejection sampling mechanism based on semantic similarity between original and reconstructed argumentative texts. We compute BERT-based F1 similarity scores between the original argumentative text $a_i^{(t)}$ and its reconstruction $\tilde{a}_{i,j}^{(t)}$:

$$\text{sim}(a_i^{(t)}, \tilde{a}_{i,j}^{(t)}) = \text{BERT-F1}(a_i^{(t)}, \tilde{a}_{i,j}^{(t)}) \tag{3}$$

We select the top 25% of summary-article pairs based on their reconstruction similarity scores:

$$\mathcal{D}_{selected}^{(t)} = \text{top}_{0.25}\{(a_i^{(t)}, s_{i,j}^{(t)}) : \text{sim}(a_i^{(t)}, \tilde{a}_{i,j}^{(t)})\} \tag{4}$$

This filtering mechanism ensures that selected summaries preserve the core argumentative content and logic of the original texts, enhancing their ability to faithfully reconstruct the source material.

#### 3.2.3 ITERATIVE FINE-TUNING PROCESS

The filtered dataset $\mathcal{D}_{selected}^{(t)}$ is used to fine-tune the model using supervised fine-tuning. The fine-tuning objective minimizes the cross-entropy loss between the model's predicted summary tokens and the ground truth summaries:

$$\mathcal{L}^{(t)} = - \sum_{(a,s) \in \mathcal{D}_{selected}^{(t)}} \sum_{i=1}^{|s|} \log P_{\theta^{(t)}}(s_i | a, s_{<i}) \tag{5}$$

After fine-tuning, the updated model parameters $\theta^{(t+1)}$ are used for the next iteration. The expanded texts from the current iteration become the input corpus for the subsequent iteration:

$$\mathcal{A}^{(t+1)} = \{\tilde{a}_{i,j}^{(t)} : (a_i^{(t)}, s_{i,j}^{(t)}) \in \mathcal{D}_{selected}^{(t)}\} \tag{6}$$

The motivation of such a process is to utilize the expanded summary as a source for more arguments in order to mitigate overfitting when the model is repeatedly fine-tuned on the same training dataset.

### 3.2.4 ALGORITHM OVERVIEW

Algorithm 1 presents the complete iterative rejection sampling framework:

---

**Algorithm 1** Iterative Rejection Sampling for Argumentative Summarization

---

**Require:** Initial dataset $\mathcal{A}^{(0)}$, model $f_{\theta^{(0)}}$, iterations $T$, sample size $k$
**Ensure:** Fine-tuned model $f_{\theta^{(T)}}$
1: **for** $t = 0$ **to** $T - 1$ **do**
2:     *// Generation Phase*
3:     **for each** $a_i^{(t)} \in \mathcal{A}^{(t)}$ **do**
4:         Generate $k$ summaries: $S_i^{(t)} = \{f_{\theta^{(t)}}(a_i^{(t)})\}_{j=1}^{k}$
5:         Reconstruct: $\tilde{A}_i^{(t)} = \{g_{\theta^{(t)}}(s_{i,j}^{(t)})\}_{j=1}^{k}$
6:     **end for**
7:     *// Filtering Phase*
8:     Compute similarity scores: $\text{sim}(a_i^{(t)}, \tilde{a}_{i,j}^{(t)})$ for all pairs
9:     Select top 25%: $\mathcal{D}_{selected}^{(t)} = \text{top}_{0.25}(\text{sim})$
10:    *// Training Phase*
11:    Fine-tune $f_{\theta^{(t)}}$ on $\mathcal{D}_{selected}^{(t)}$ to obtain $f_{\theta^{(t+1)}}$
12:    Update corpus: $\mathcal{A}^{(t+1)} = \{\tilde{a} : (a, s) \in \mathcal{D}_{selected}^{(t)}\}$
13: **end for**
14: **return** $f_{\theta^{(T)}}$

---

## 4 EVALUATION AND RESULTS

### 4.1 EXPERIMENT SETTINGS

We employ a learning rate of $\eta = 1 \times 10^{-5}$ with the AdamW optimizer using 8-bit quantization for memory efficiency. Training uses a batch size of 4 per device with gradient accumulation steps of 2, effectively yielding a batch size of 8. Each iteration consists of 2 training epochs with 5 warmup steps and weight decay $\lambda = 0.01$ in the fine-tuning phase. We have a total of 4 iterations for each LLM. The loading and fine-tuning of LLMs use a fixed random seed of 3407 for reproducibility. We implement our framework using LoRA (Low-Rank Adaptation) for efficient fine-tuning, with parameters $r = 16$, $\alpha = 16$, targeting projection matrices in attention and feed-forward layers. The maximum sequence length is set to 2048 tokens with a maximum generation length of 1024 tokens.

For computational efficiency, we process data in batches and implement memory management techniques including gradient checkpointing and 4-bit quantization. The framework runs for 8 iterations on each dataset, with early stopping if no suitable training examples are found in any iteration. The BERT scoring model used is DeBERTa-large-MNLI from the DeBERTa family (He et al., 2021).

### 4.2 RESULTS AND ANALYSIS

As shown in Table 1, during the iterative process, the score had improvements from 0.84% up to 11.88%, even though the score might also trend down as a sign of overfitting outside of the Gemma experiments. The Gemma experiments exhibit a different pattern of overfitting from the beginning, which likely stems from overfitting and requires finding better hyper-parameters, like learning rates.

The larger, commercially available LLMs perform better on this task than the models we have fine-tuned. However, the BERT F1 score is still low, indicating the difficulty of this task. In addition, Qwen/Qwen3-4B-Instruct-2507 performs the best across iterations, with an average F1 score of 0.121 (not shown in Table 1) but reaching a plateau, as it might have been aligned to such tasks and domains before fine-tuning, while Llama-3.2-3B has the largest proportional performance gain.

Therefore, we will further analyze the trajectory of Llama-3.2-3B and especially as compared to 1B. Since the increase of length of the summary is not penalized, LLMs might increase the token length of the summarization between iterations to achieve a better score measured by this framework. However, as shown in Figure 2, Llama-3.2-3B and Llama-3.2-1B perform differently on the

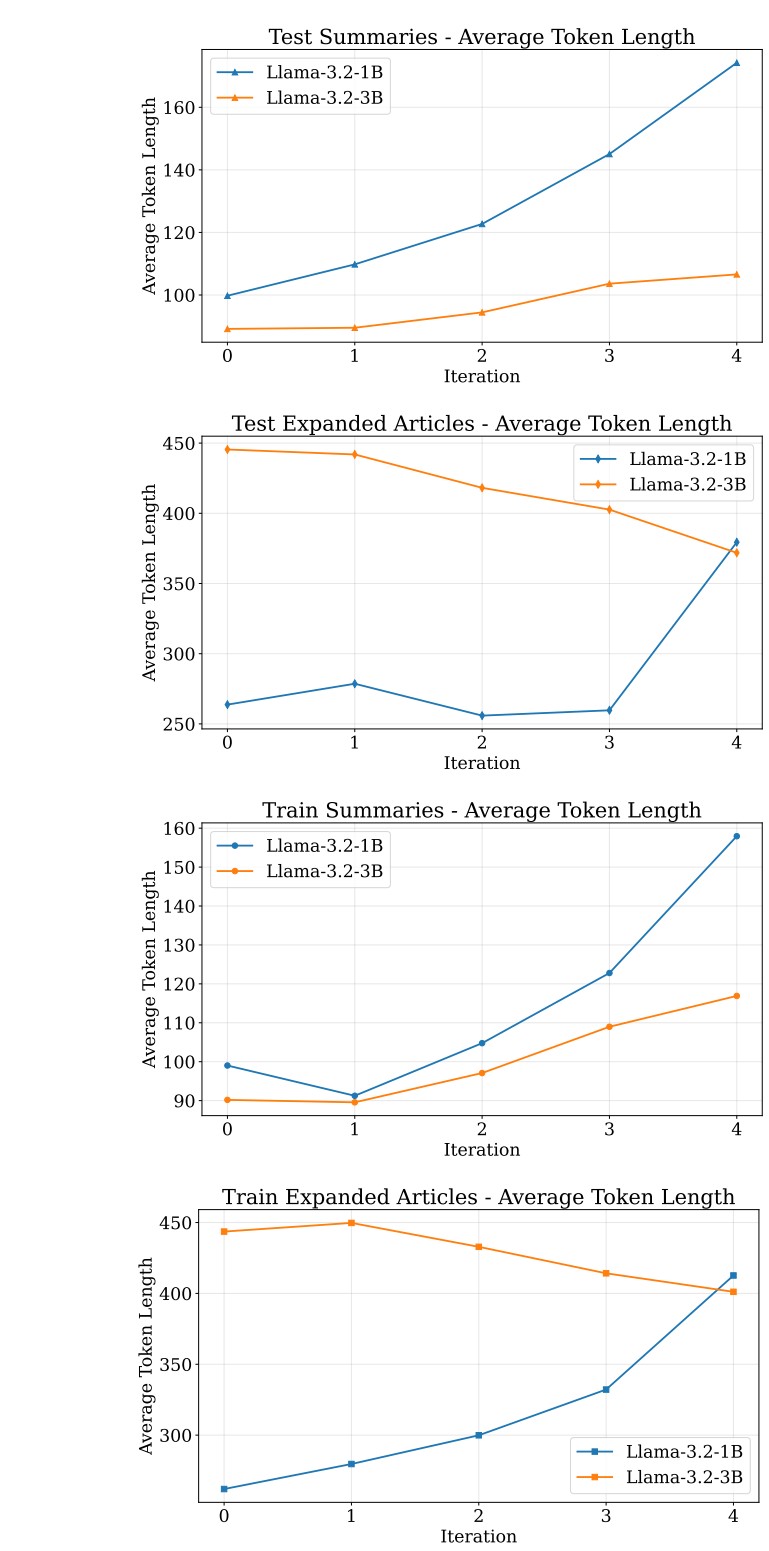

Figure 2: Visualizations on how Llama-3.2-3B and Llama-3.2-1B behaves differently at average summarization and expansion token numbers. While Llama-3.2-3B outperforms Llama-3.2-1B in terms of performance gains, it has a more moderate increase on summarization compared to Llama-3.2-1B, and actually exhibits a decrease on the expanded summary, indicating that the knowledge on summarization of the arguments might have transferred onto the expansion task.

Table 1: Average BERT-F1 scores across the test set. The Gemma experiments are cut-off as they have overfitted and would require changes to the hyper-parameters, such as the learning rate.

| Model | Iterations | | | | |
|---|---|---|---|---|---|
| | 0 | 1 | 2 | 3 | 4 |
| meta-llama/Llama-3.2-1B-Instruct | 0.064 | 0.073 | 0.068 | 0.057 | 0.041 |
| meta-llama/Llama-3.2-3B-Instruct | 0.101 | 0.102 | 0.105 | 0.112 | 0.113 |
| Google/Gemma-3-1B-it | 0.090 | 0.089 | 0.089 | | |
| Google/Gemma-3-4B-it | 0.103 | 0.098 | 0.097 | | |
| Qwen/Qwen3-4B-Instruct-2507 | 0.119 | **0.118** | **0.120** | **0.119** | **0.120** |
| Deepseek/chat (DeepSeek-V3.1) | **0.123** | | | | |
| OpenAI/gpt-4o-mini | 0.105 | | | | |
| Claude/sonnet-4-20250514 | 0.120 | | | | |

summarization task. For summaries (top row), both the training and test sets show a steady increase in length as iterations progress. The growth is particularly pronounced for the 1B model, which increases from around 100 tokens to nearly 160, while the 3B model increases more moderately and stabilizes around 90–115 tokens. This suggests that smaller models are more prone to "length inflation" in iterative summarization, whereas larger models remain more stable. For expanded articles (bottom row), the two models diverge. The 1B model consistently produces longer outputs with each round, increasing from roughly 260 to more than 400 tokens. In contrast, the 3B model starts with longer generations ($\sim$ 450 tokens) but gradually contracts to around 390 tokens as iterations proceed. This might indicate that in the case of Llama-3.2-3B, there might be a more significant and effective learning of the summarization of arguments that is transferred to the task of expansion, and the model also adapted better the summarization task than Llama-3.2-1B.

## 5 CONCLUSION

In this work, we introduced an iterative rejection sampling framework that addresses the significant challenges Large Language Models (LLMs) face when summarizing argumentative text. Our approach leverages a self-supervised cyclical process that generates high-quality argument-summary pairs without requiring external human annotation, addressing the fundamental problem of domain adaptation for argumentative discourse.

Our key contributions include: (1) a novel reconstruction-based quality assessment mechanism that uses semantic similarity between original and reconstructed arguments to filter training data, (2) an iterative fine-tuning framework that progressively improves model performance through self-generated examples, and (3) comprehensive evaluation demonstrating improvements up to 11.88% in BERT F1 similarity scores on the r/ChangeMyView dataset.

The experimental results reveal important insights into model behavior during iterative training. While our framework consistently improves performance for larger models like Llama-3.2-3B, we observed that smaller models (Llama-3.2-1B) are more susceptible to length inflation, producing increasingly verbose outputs that may compromise the core objective of concise summarization. This suggests that model capacity plays a crucial role in maintaining the balance between comprehensiveness and brevity in argumentative summarization.

Our analysis of token length dynamics shows that the Llama-3.2-3B model not only outperforms smaller variants but also demonstrates more stable behavior, even showing contraction in expanded text length over iterations. This indicates that larger models may be better at transferring learned summarization capabilities to related tasks like argument expansion, suggesting a more robust understanding of argumentative structure.

The framework's self-supervised nature makes it particularly valuable for domains where high-quality training data is scarce. By eliminating the need for human-annotated argument-summary pairs, our approach enables scalable adaptation to specialized argumentative domains while maintaining quality through reconstruction-based filtering.

**Limitations and Future Work.** Despite these promising results, several limitations warrant attention. The framework's tendency toward length inflation in smaller models suggests the need

for explicit length constraints or penalties in the training objective. Additionally, our evaluation is currently limited to a single dataset (r/ChangeMyView), and broader evaluation across diverse argumentative domains would strengthen the generalizability claims.

Future research directions include investigating alternative quality metrics beyond reconstruction similarity, exploring the framework's applicability to other specialized domains requiring logical reasoning, and developing methods to mitigate length inflation while preserving content quality. The integration of explicit logical structure preservation mechanisms could further enhance the framework's ability to maintain argumentative coherence.

Our work demonstrates that self-supervised iterative training can meaningfully improve argumentative summarization capabilities in LLMs, providing a scalable path toward better domain adaptation for complex reasoning tasks. Our framework is characterized by its use of a pair of tasks that counter each other out, summarization and expansion, to obtain a self-supervised signal for iterative fine-tuning. The same concept should be applicable to a broader range of such task pairs, such as creating and solving programming problems.

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

## 6 APPENDIX

### 6.1 ETHICS STATEMENT

This work investigates iterative summarization and expansion of argumentative text using Large Language Models (LLMs). We primarily rely on the publicly available Reddit ChangeMyView dataset following the license (CC-BY-4.0). This dataset has been widely adopted in prior research

on argumentation and reasoning. To minimize risks to privacy, we use only de-identified text that has already been released for research purposes, and no attempt is made to deanonymize or trace content back to individual users.

The models used in our experiments include both open-source LLMs (via Unsloth) and hosted APIs (Anthropic Claude, OpenAI GPT, DeepSeek). We follow the provider's terms of service and no sensitive or private data are submitted to the APIs. Generated outputs are used strictly for research evaluation and not for deployment in real-world decision-making systems.

Potential ethical concerns include the amplification of biases present in the training data and the possibility of producing harmful or misleading content. To mitigate these risks, we limit our experiments to summarization and expansion tasks within a controlled research setting and only evaluation metrics such as BERTScore for quality control. The results are not intended for direct downstream applications where misinformation or bias could cause harm.

Our study complies with the the principles of research integrity: we document model configurations, hyperparameters, and evaluation protocols to ensure reproducibility. In this study, no human subjects, private information, or sensitive demographic characteristics are involved and therefore IRB approval was not required.

## 6.2 LARGE LANGUAGE MODELS USAGE

The background information, the ideation process, and the writing of this paper were done with the aid of LLMs capable of searching for information online. The details of the algorithm have been modified and optimized with information from LLMs, and the writing was also improved using LLMs.

