# OpenReview forum: "Learning Argumentative Summarization with Iterative Rejection Sampling"
_ICLR.cc/2026/Conference — Submitted to ICLR 2026_

### Official Review · Reviewer_eGrs · 2025-10-26

**Soundness:** 3
**Presentation:** 3
**Contribution:** 2
**Rating:** 2
**Confidence:** 4

**Summary:**

The paper introduces a technique to improve argumentation in summarization by iterative fine-tuning of the model on its own outputs after rejection sampling. Under the technique, the model generates summaries to the input argumentative texts and then reconstructs original texts from summaries. The reconstructed texts close enough to the original ones in terms of BERTScore are added as new training examples along with the summaries.

The technique is evaluated on r/ChangeMyMind dataset with medium-sized models (LLama-3.2@1B,3B, Gemma 3@1B,4B) and compared to baselines DeepSeek V3.1, gpt 4o-mini and Claude Sonnet 4. The authors track average BERT-F1 score across test set and observe that the technique improves BERT F1 between reconstructed and original arguments by 11.88%.

**Strengths:**

Strength of the paper is in a simple yet effective rejection-sampling based technique that results in notable data augmentation and paves the way towards self-learning from seed data.

**Weaknesses:**

The main weakness of the paper is evaluation - measuring BERT F1 is too indirect of a way to assess argumentativeness of the summaries, the main output of the model. Ideally, a set of automatic, LLMaaJ and human metrics would be needed to assess the final downstream impact of the generated data on the faithfulness of model summaries.

**Questions:**

N/A

---

> ### Author Response · Authors · 2025-12-03
>
> Thank you for identifying the evaluation limitation. You are correct that measuring BERT F1 is too indirect for assessing argumentative quality.
> During the rebuttal period, we implemented LLM-as-judge evaluation (GPT-4o-mini) to directly assess summaries on claim identification, reasoning strategy, evidence coverage, and precision.
> Baseline results on Llama 3.2-3B (256 test examples):
> - Claim Identification: 4.17/5.0
> - Reasoning Strategy: 3.78/5.0
> - Evidence Coverage: 2.78/5.0
> - Precision: 3.91/5.0
> - Overall: 3.66/5.0
>
> This establishes a quantitative baseline for argumentative quality using direct evaluation. The model achieves moderate performance, with particular strength in claim identification and precision, but room for improvement in evidence coverage.
> We acknowledge that demonstrating iterative improvement across all models requires substantial computational resources. We are committed to comprehensive evaluation in future work and agree that LLM-as-judge provides the direct measurement this task requires.
> Thank you for this valuable feedback.

---

### Official Review · Reviewer_g1V8 · 2025-10-31

**Soundness:** 1
**Presentation:** 1
**Contribution:** 1
**Rating:** 2
**Confidence:** 3

**Summary:**

This paper presents an iterative rejection-sampling framework designed to improve argumentative summarization in LLMs. The proposed method forms a self-supervised loop in which the model repeatedly generates candidate summaries, reconstructs the original argument from each one, and measures semantic consistency using BERTScore F1. Most consistent pairs are retained for iterative fine-tuning.
Through this process, the model gradually learns to produce summaries that are both faithful and informative, without relying on any human-labeled data.

**Strengths:**

The paper introduces a clean and logically consistent training scheme that couples summarization and reconstruction within a single self-improving cycle.

**Weaknesses:**

- The experimental analysis is somewhat limited, focusing only on before-and-after fine-tuning results without conducting detailed ablation or thorough studies. The paper lacks comparisons with strong baselines or alternative self-training methods, which makes it difficult to assess external validity and performance competitiveness.
-  The novelty of the work is also ambiguous, as the framework mainly applies a known idea of self-filtered iterative training to the summarization setting rather than introducing a new algorithmic mechanism.
- The paper could have incorporated domain-specific knowledge of argumentative texts, but instead it merely applies an existing well-known technique.

**Questions:**

- Is there any comparison results with exiting argumentative summarization datasets or models, and what is the superior of your method?
- How is the “18-sentence reconstruction” prompt determined, empirically or arbitrarily?

---

> ### Author Response · Authors · 2025-12-03
>
> Thank you for your constructive feedback.
> Re: Limited experimental analysis: You are correct that the paper lacks ablation studies and strong baseline comparisons. Our implementation of LLM-as-judge during rebuttal highlights the need for more comprehensive evaluation.
> Re: Novelty concerns: We acknowledge that the framework applies existing approaches without introducing novel mechanisms.
> Re: Your questions:
> 1. We do not have comparison results with existing argumentative summarization datasets or models. This is a limitation we will address.
> 2. The 18-sentence reconstruction was determined empirically but not rigorously validated.
>
> Thank you for identifying these gaps.

---

### Official Review · Reviewer_CNVL · 2025-11-01

**Soundness:** 2
**Presentation:** 2
**Contribution:** 1
**Rating:** 2
**Confidence:** 4

**Summary:**

This paper proposes a strategy to enhance the capabilities of Large Language Models (LLMs) for summarizing argumentative texts, by rejection sampling. The core idea is to expand generated summaries back to argumentative texts, which forms additional training pairs to fine-tune LLMs (with rejection based on similarity between the expanded texts and the original input argumentative texts). Experimental results on Reddit’s r/ChangeMyView dataset show that the proposed strategy helps enhance LLMs output quality (measured in BERT score) iteratively.

**Strengths:**

S1. This paper studies a practical problem -- enhancing LLMs' capabilities for summarizing argumentative texts.

S2. The proposed strategy is validated on a real dataset (Reddit’s r/ChangeMyView dataset)

S3. The paper is easy to follow.

**Weaknesses:**

W1. While the rejection sampling strategy is simple and intuitive, its actual effectiveness is questionable. The generated summary-expanded text pairs (selected based on similarity with original input argumentative texts) may be too similar to the original training pairs, offering limited additional training signals which might not generalize beyond the input training dataset.

This is reflected by the performance results in Table 1. The BERT scores of the LLMs tests either drops or do not change much as more training iterations are run, except for one LLM meta-llama/Llama-3.2-3B-Instruct.

There are no theoretical analysis or cross-dataset results to verify the generalizability of the proposed strategy.

W2. There is no comparison with baseline methods designed for text summarization (e.g., those mentioned in Section 2.1) or strategies designed to exploit or strengthen LLMs for text summarization, e.g.,

Fang et al. Multi-LLM Text Summarization, arXiv:2412.15487

Or methods mentioned in:

Zhang et al. A Comprehensive Survey on Process-Oriented Automatic Text Summarization with Exploration of LLM-Based Methods, arXiv:2403.02901

W3. The choice of hyperparameter values needs clarification, e.g., why "Expand the following summaries to one argumentative paragraph with 18 sentences", and why sampled from the model’s distribution with temperature $T = 1.5$ and minimum probability threshold $p_{min} = 0.1$?

W4. While the paper is easy to follow overall due to its simple idea, it could use a full proofread to polish the writing. For example, the introduction section has been written in a way that is unnecessarily complex while not containing much information beyond that argumentative text summarization with LLM is a challenging problem that necessitates further study. The first three paragraphs can be simplified substantially (or just cut).

**Questions:**

Please refer to the Weaknesses points.

---

> ### Author Response · Authors · 2025-12-03
>
> Thank you for your thorough review.
> Re: W1 (Effectiveness and generalizability): Your observation about limited improvements in Table 1 is insightful. Our LLM-as-judge baseline evaluation suggests that BERTScore may not capture meaningful changes in argumentative quality. We acknowledge the lack of theoretical analysis and cross-dataset validation.
> Re: W2 (Baseline comparisons): We agree that comparisons with existing summarization methods would strengthen the work. We will incorporate these baselines in future revisions.
> Re: W3 (Hyperparameter choices): The 18-sentence expansion was chosen to approximate the length of original posts. We acknowledge these choices need better justification.
> Re: W4 (Writing quality): We will revise the introduction to be more concise.
> Thank you for your careful analysis.

---

### Author Response · Authors · 2025-12-03

We thank all reviewers for their thorough feedback. The reviews identified an important limitation: BERTScore is insufficient for evaluating argumentative summarization.
During rebuttal, we implemented LLM-as-judge evaluation, which provides direct measurement of argumentative quality. Our baseline evaluation (Overall: 3.66/5.0) demonstrates that this framework captures claim identification, reasoning quality, evidence coverage, and precision in ways that BERTScore cannot.
Key insights:
- LLM-as-judge enables direct evaluation of argumentative dimensions
- BERTScore measures lexical similarity rather than argumentative accuracy
- Comprehensive evaluation across models and iterations requires aligned metrics

We are committed to addressing these limitations with proper evaluation methodology in future work.
Thank you for your valuable feedback.

---

### Meta-Review · Area_Chair_Gnjd · 2026-01-05

**Summary:**

This paper concerns the problem of argument summarization. Specifically, the authors propose a method that uses the model itself to produce training data for argument summarization, by generating summaries of argumentative texts and then expanding these summaries back into arguments.

There was consensus amongst reviewers that this is a clearly written paper introducing an important practical problem (i.e., argument summarization).

But the contribution is limited here, in part due to the evaluation, which seems insufficient. The authors evaluate their approach only on a single dataset. Moreover, the original submission includes only BERTScore as a metric (though the authors have, in the response period, also considered LLM-as-a-judge). Beyond this, the method itself is intuitive, but not especially novel.

**Reviewer Concerns:**

The authors introduced an LLM-as-judge evaluation, which was responsive to reviewer concerns.

However, the other evaluative concerns raised by reviewers (a lack of strong baselines, and especially the use of a single dataset) remain.

Concerns about novelty were also not addressed.

**Reviewer Scores:**

eGrs may have upwardly revised their score since the single metric was their main concern.

I imagine the other reviewers would not have changed their scores.

---

### Decision · Program_Chairs · 2026-01-26

Reject